# Climate Change and Vector-Borne Diseases in China: A Review of Evidence and Implications for Risk Management

**DOI:** 10.3390/biology11030370

**Published:** 2022-02-25

**Authors:** Yurong Wu, Cunrui Huang

**Affiliations:** 1Vanke School of Public Health, Tsinghua University, Beijing 100084, China; wuyr26@mail2.sysu.edu.cn; 2School of Public Health, Sun Yat-sen University, Guangzhou 510275, China; 3Institute of Healthy China, Tsinghua University, Beijing 100084, China

**Keywords:** climate change, meteorological factor, vector-borne disease, regional differentiation

## Abstract

**Simple Summary:**

Vector-borne diseases are among the most rapidly spreading infectious diseases and are widespread all around the world. In China, many types of vector-borne diseases have been prevalent in different regions, which is a serious public health problem with significant association with meteorological factors and weather events. Under the background of current severe climate change, the outbreaks and transmission of vector-borne diseases have been proven to be impacted greatly due to rapidly changing weather conditions. This study summarizes research progress on the association between climate conditions and all types of vector-borne diseases in China. A total of seven insect-borne diseases, two rodent-borne diseases, and a snail-borne disease were included, among which dengue fever is the most concerning mosquito-borne disease. Temperature, rainfall, and humidity have the most significant effect on vector-borne disease transmission, while the association between weather conditions and vector-borne diseases shows vast differences in China. We also make suggestions about future research based on a review of current studies.

**Abstract:**

Vector-borne diseases have posed a heavy threat to public health, especially in the context of climate change. Currently, there is no comprehensive review of the impact of meteorological factors on all types of vector-borne diseases in China. Through a systematic review of literature between 2000 and 2021, this study summarizes the relationship between climate factors and vector-borne diseases and potential mechanisms of climate change affecting vector-borne diseases. It further examines the regional differences of climate impact. A total of 131 studies in both Chinese and English on 10 vector-borne diseases were included. The number of publications on mosquito-borne diseases is the largest and is increasing, while the number of studies on rodent-borne diseases has been decreasing in the past two decades. Temperature, precipitation, and humidity are the main parameters contributing to the transmission of vector-borne diseases. Both the association and mechanism show vast differences between northern and southern China resulting from nature and social factors. We recommend that more future research should focus on the effect of meteorological factors on mosquito-borne diseases in the era of climate change. Such information will be crucial in facilitating a multi-sectorial response to climate-sensitive diseases in China.

## 1. Introduction

Except for COVID-19, climate change may be the most important global public health issue of our time [1] since it threatens all aspects of human health, including the risks of infectious diseases [2]. Vector-borne disease is one of the infectious diseases most sensitive to climate. It has put 80% of the population at infection risk [3] and causes more than 700,000 deaths every year all around the world [4]. Unfortunately, climatic conditions, especially rising temperature, have also been contributing to outbreaks of vector-borne diseases in recent years [5]. Previous studies have predicted that the spreading areas of vector-borne diseases will expand in many countries due to a rapidly changing climate [6,7]. Climate change has been proven to be one of the key drivers of changing the life activities, living environments, migration, and geographical distribution of vectors [8,9], which will ultimately increase the risk of vector-borne diseases [10].

China is one of the most vulnerable countries with respect to climate change [11]. Many studies have found the significant positive relationship between weather conditions and vector-borne diseases in China, including the effects of meteorological factors, such as temperature, precipitation, humidity, wind speed [12,13,14,15,16,17], and extreme weather events (e.g., extremely high temperatures, extremely high rainfall, and tropical cyclones) [18,19,20,21,22,23,24]. The 2020 China Report of the Lancet Countdown on Health and Climate Change showed that China’s climate suitability for mosquito-borne dengue fever has increased by 37% in the past half century [25]. If nothing is done to address climate change, the malaria incidence in northern China will rise from 69% to 182% by 2050 [26].

There are many types of vector-borne diseases prevalent, and the relationship between climate conditions and vector-borne diseases can be completely different among regions [27]. However, most researchers have conducted their studies in one province or one climate zone in China [28,29], and previous reviews only summarized the research progress of a certain type of vector-borne disease, such as mosquito, insect, or rodent [30,31,32]. In terms of the Chinese studies in this field, the scale of research has not been adequately understood, and the overall impact of climate conditions on vector-borne diseases transmission and distribution has been poorly investigated. It is necessary to systematically review the research progress of how changes in weather patterns may impact climate-sensitive vector-borne diseases in China.

The purpose of this paper is to review all relevant studies concerning weather, climate, and vector-borne diseases to identify the most sensitive climate-related vector-borne diseases in China. In addition, we investigated how these meteorological factors impact vector-borne diseases and regional differences in their relationships. We also analyze the limitations of the current studies and put forward some suggestions for future research.

## 2. Methods

### 2.1. Search Strategy

Considering that there are few publications before the year 2000, we limited the time of publication, assessing the effects of meteorological factors or climate events on vector-borne diseases between January 2000 and December 2021. In order to comprehensively retrieve all relevant studies in China, the current systematic review was undertaken using seven Chinese and English databases, including PubMed, Embase, Scoups, CBM, CNKI, Wanfang Database, and VIP.

The search strategy included exposure keywords, such as “climate change”, “climate factors”, “meteorological factors”, “environmental factors”, “extreme weather events”, and “weather”, and outcome keywords included “vector-borne disease”, “mosquito-borne disease”, and “rodent-borne disease”. We then included “temperature”, “precipitation OR rainfall”, and “humidity” into exposure keywords due to the high frequency of studies on them. “Dengue”, “malaria”, and “schistosomiasis” were included as outcome keywords because of their high prevalence and number of studies shown in preliminary search results. In the process of reading the full text, the references that met the eligibility criteria were also included.

### 2.2. Selection Criteria

We included empirical research with available full-text published in peer-reviewed journals, excluding dissertations, conference proceedings, reviews, and commentary papers. Due to the large difference in public health management between Hong Kong, Macao, Taiwan, and mainland China, the scope of our study was restricted to mainland China. Only epidemiological studies were reviewed, and laboratory studies on viruses carried by vectors, population immunity, or genes and studied only on animals or vectors were excluded.

We reviewed all climate-related factors, including meteorological factors and extreme climate events, and all types of health outcome index with no population or method limits in order to summarize comprehensively the research progress.

### 2.3. Study Selection Process

Literature retrieved from the seven Chinese and English databases were imported into the Endnote X9 reference management system, and duplicates were subsequently removed. All retrieved studies were then identified by the initial screening of titles, followed by abstract screening. If the information in titles or abstracts was not sufficient to decide on the inclusion or exclusion of the study, then the full text was reviewed for eligibility. The studies were screened against inclusion/exclusion criteria by two authors independently. Inconsistencies between the two authors were discussed for clarification and agreement on final reporting with the other authors. References in each of the identified papers were also examined for any additional studies that may have been missed in our database searches. The study selection process is shown in Figure 1.

### 2.4. Data Extraction and Analysis

Relevant study characteristics were extracted from eligible studies into a Microsoft Excel spreadsheet. The characteristics included author; journal; year; vector-borne disease; vectors; research objectives; region; time period; exposure metrics; source of exposure metrics; outcome metrics (morbidity, density of morbidity, etc.); source of outcome metrics; method; main results; and effect path of meteorological factors or climate events on vector-borne diseases. We used R studio 4.1.0, Microsoft Excel to visualize the geographical and temporal distribution of eligible studies. Study results reported positive, negative, or another correlation between temperature, precipitation, and humidity, and vector-borne diseases were extracted and listed.

## 3. Results

### 3.1. Literature Description

Our search retrieved a total of 3149 records. After the process of literature screening according to eligible criteria, a total of 131 articles were included for full-text screening and data extraction, including 30 Chinese articles and 101 English articles, respectively.

A total of ten types of vector-borne disease were included: seven types of insect-borne disease (malaria, dengue, and Japanese encephalitis transmitted by mosquito; scrub typhus transmitted by mites; high fever with thrombocytopenia syndrome transmitted by ticks; typhus group rickettsiosis transmitted by fleas; and visceral leishmaniasis transmitted by sandflies), two types of rodent-borne diseases (plague and hemorrhagic fever with renal syndrome), and one snail-borne disease (schistosomiasis). Figure 2a shows the temporal tendency of the literature both in Chinese and in English between 2000 and 2021. The number of studies increased largely after the year 2010 due to growing concerns about climate change [10]. Figure 2b shows the number of published studies in Chinese and English on different vector-borne diseases. Insect-borne diseases, especially mosquito-borne diseases, have the highest number of studies.

Figure 2c shows the geographic distribution of publications in China, with the highest number in Guangdong Province followed by Yunnan Province, Anhui Province, and Shandong Province. The studies of insect-borne diseases were mostly conducted in the areas of central and southern areas in China [33,34,35,36,37,38,39]. Among them, malaria studies were mostly in counties [40,41], while dengue studies were concentrated in the southern coastal areas, with Guangzhou being the province with the highest number of studies. The research areas of rodent-borne diseases, hemorrhagic fever with renal syndrome epidemics, and plague were mostly distributed in the northern and northwest regions of China [42,43,44,45]. Schistosomiasis was mostly studied in the central part of China with rich water bodies, including the Yangtze River basin, Hanjiang River basin, and Poyang Lake region [46,47,48,49]. 

### 3.2. The Relationship between Meteorological Factors and Vector-Borne Diseases

The characteristics of the reviewed studies are shown in Table 1. Most published studies used ecological design for risk assessment, in which the most frequently used mathematical models were the Ecological Niche Model (ENM), Distributed Lag Non-linear Model (DLNM), Generalized Estimated Equation (GEE), Generalized Linear Model (GAM), Logistic Regression and Autoregressive Integrated Moving Average Model (ARIMA), etc. The relationship between climate factors and vector-borne diseases was non-linear; a J-shape or reverse U-shape was always found between them, which means, respectively, that the risks increased continuously or increased and then decreased with the rise of certain meteorological factors. Temperature, precipitation, and humidity were the main parameters contributing to the transmission of vector-borne diseases [13,50,51,52]. Generally, temperature plays an important role in the number of reported insect-borne and rodent-borne diseases cases [42,53,54,55,56]. However, typhus group rickettsiosis was negatively related to average temperature, average ground temperature, and extreme minimum temperature [13]. The same negative correlation was also found between temperature and schistosomiasis [47]. Precipitation can promote the transmission of most insect-borne diseases [57,58] and interacts with temperature [16]. It is expectable that moderate precipitation (10–120 mm) and temperatures of 10–25 °C were the most favorable condition for HFRS incidence [59]. In addition, rising humidity is facilitates insect-borne disease transmission, such as dengue and plague [60,61]. Atmospheric pressure and wind speed were inversely related to vector-borne diseases but sunshine was positively related [52,62,63].

A few studies have assessed the effect of climate events on vector-borne diseases. A precipitation pattern with a cumulative precipitation of 20.67–55.50 mm per week, 3–4 days per week with light or moderate precipitation, and a coefficient of precipitation variation of less than 1.41 per week might be an optimal precipitation pattern for dengue epidemics in Guangzhou [64]. Another study conducted in Mengcheng County showed that an increased risk of malaria was significantly associated with flooding and waterlogging [65]. Dengue and hemorrhagic fever with renal syndrome were inversely related to the Southern Oscillation Index [66,67] but plague was positively related to it [68]. Another study found that the relationship between the East Asian monsoon index and dengue was reverse-U shaped [69]. In addition, the outcome metrics of most studies were incidence or number of cases in all studies, while Xu found humidity was an effective factor on the intensity and transmission speed of plague [61,70].

### 3.3. Potential Pathway of Meteorological Factors on Vector-Borne Diseases

Figure 3 shows the main effect path of the climate change on vector-borne diseases. Climate change can contribute to a lot of environmental problems, such as changing vegetation cover and land type, accelerating the melting of snow, and exacerbating the urban heat island effect, impacting urban water supply systems and population mobility. These changes can lead to the disruption of ecosystems’ balance and loss of wildlife habitats, which may affect the reproduction, survival, spread, and distribution of pathogens, vectors, or intermediate hosts, ultimately increasing the risk of vector-borne diseases [71].

Rising temperatures can shorten the incubation and reproduction rates of dengue viruses and the life cycle of mosquitoes, which can contribute to the increase in the number and spread rate of vectors [66,72,73]. Moreover, high temperatures can increase vectors’ survival rate [74], but when the temperature exceeds a certain threshold, it has an adverse effect on the reproduction and bite rates of insects [75]. When the temperature continues to be above 37 °C for several days, the mosquito breeding grounds are reduced by the evaporation of water, which in turn leads to a decrease in mosquito density and affects the prevalence of malaria [76]. Snails are the vectors of schistosomiasis and cannot survive in areas where temperatures are below 0 °C in January. Studies have proven that the habitat suitable for snails in Poyang Lake has moved northward due to rising temperatures [48,77,78]. In addition, rising soil surface temperatures can significantly affect the transmission of leishmaniasis by impacting the growth and development of its vector, sandflies, which carry out life activities in the first three life stages in soil close to the surface [79].

Precipitation can provide more habitats for mosquitoes, contributing to their survival and reproduction [80,81]. However, extreme precipitation can destroy vector habitats, disrupt the growth of insects, and wash eggs out of breeding grounds, further decreasing vector density and disease transmission [81,82,83]. Some scholars also believe that although heavy precipitation takes away vector organisms, the rest of the rain will become a potential breeding ground for adult mosquitoes [84]. In addition, heavy rainfall in eastern China also destroys rodent habitats, reducing rodent–rodent contact, human–rodent contact, and the spread of the virus [18]. However, in southern China, the migration of infected people and rodents may promote the spread of plague between regions, although flooding can kill the vectors [61].

### 3.4. The Regional Differentiation of the Relationship between Meteorological Factors and Vector-Borne Diseases

Both the relationship and the mechanism mentioned above have shown large regional differences. The correlation between the main meteorological factors (temperature, precipitation and humidity) and vector-borne diseases in different provincial/municipal/county administrative regions are shown in Table 2, in which “+” represents a positive correlation, “−” represents a negative correlation, “J” represents a J-shaped correlation, and “reverse U” represents a reverse U-shaped correlation. Generally, in southern China, temperature may contribute to the transmission of insect-borne diseases while adversely affecting it in the northern region. In contrast, the number of rodent-borne diseases cases may be decreased by temperature in southern China but be increased in the northern region. Meanwhile, a positive association between precipitation and insect-borne diseases can be found in the southern China, while the same relationship between precipitation and rodent-borne diseases is mainly distributed in the northern region.

The pathway of climate change impacting vector-borne diseases can also vary among regions. Small increases in temperature can greatly affect the spread of malaria in areas with lower average temperatures since the prevalence of malaria in hotter regions is much higher than in colder regions [85]. In addition, rainwater does not gather readily in the Yunnan–Guizhou Plateau because of the typical mountainous characteristics of it, and currents can destroy mosquito breeding habitats and reduce mosquito population density, which may not lead to the incidence of JE increasing with increases in rainfall in this area [86]. Heavy precipitation can also destroy rodents’ habitats and reduce its populations. However, due to low winter temperatures in northern China, heavy precipitation may cause rodents to gather indoors, increasing the likelihood of human–rodent contact [16,70].

## 4. Discussion

We systematically reviewed 131 studies that investigated the association between climate conditions and vector-borne diseases, specifically referring to insect-borne diseases, rodent-borne diseases, and snail-borne diseases in China, with the evidence gathered covering all the regional blocks.

The most significant health threat faced from climate-related vector-borne diseases in China is mosquito-borne diseases, while the concern and health threat from rodent-borne diseases is decreasing. Aedes aegypti and Aedes albopictus are the main mosquito vectors transmitting viruses. Globally, Aedes aegypti, mainly distributed in South America, play an active role in increasing Zika transmission risks. Aedes albopictus, on the other hand, is mainly distributed in the southeastern United States, southern China, and the northern summer season in southern Europe; it is the main vector of mosquito-borne diseases in China [87]. In the last few decades, the mosquito-borne disease malaria once posed a high infection risk in China. However, on June 3th 2021, China was declared malaria-free by the World Health Organization, although today 40% of the world’s population still lives in malaria-endemic regions, with Africa being the most severe malaria-endemic region. Within the past few years, dengue fever has become the most important mosquito-borne disease health threat facing China, especially in Guangdong Province. The transmission risk of dengue is significantly increasing since it has been spreading from the coastal areas of southern China to northern areas in recent years [34]. The disease with the second highest number of publications we reviewed is schistosomiasis, which is endemic in more than 70 developing countries worldwide, but it has been well controlled in China recently [46]. Importantly, it is worth noting that China has been the most severely impacted epidemic area for renal comprehensive hemorrhagic fever in the world, with nearly 10,000 cases reported annually [16]. However, few targeted studies are currently available. It is critical to understand the impact of climate change on renal comprehensive hemorrhagic fever in the future.

Temperature, precipitation, and humidity were the main climate factors contributing to the transmission of vector-borne diseases [88,89,90]. Another review conducted in East Africa similarly found a great deal of literature on the relationship between precipitation and temperature and tropical diseases, such as dengue, chikungunya, and leishmaniasis [91]. Nevertheless, the body of evidence linking these three climatic variables was very high worldwide due to their persistence of the influence. Firstly, for the studies of temperature, the association of it with vector-borne diseases can be totally different. A study showed that limiting the global average temperature increase to 1.5–2 °C could reduce the incidence and spatial transmission of dengue in Latin America [92]. While in the Gambia region, the direct effects of increased temperatures may promote greater environmental suitability for dengue but reduce the suitability of Anopheles mosquitoes for malaria transmission [93]. Secondly, precipitation impact vector-borne diseases incidence and distribution in a complex way. Increased rainfall may alter regional ecology and thus local ecological diseases. It has been shown that the risk of dengue increased between 0 and 3 months after extremely wet conditions and between 3 and 5 months after drought conditions in Brazil [94]. In the United States, unusual El Niño-related precipitation caused a 20-fold increase in rodent populations between 1992 and 1993 [95]. Thirdly, humidity, both related to temperature and rainfall, can increase egg production, larval indices, and activities of mosquitoes. Studies conducted in Saudi Arabia have shown that a suitable range of humidity stimulating mosquito flight activity is between 44% and 69%, with the most appropriate being 65% [96].

The correlations between climate and vector-borne diseases were also hugely different among regions throughout Chinese mainland. One reason was the influence of regional factors, ecological regional factors and social regional factors included. Climatic differences between regions can be a critical contributor. For instance, it is more suitable for insects in the south of China, where it is warmer and wetter, compared with the north, so insect-borne disease more likely to occur in southern cities, such as Guangdong, Fujian, and Yunnan. In addition, the relationship between temperature and rodent-borne diseases showed different types of association, in that it was linear in the temperate zone and non-linear in the warm temperature zone [16]. Additionally, land type can impact vector habitats; for example, the typical mountainous characteristics of the Yunnan–Guizhou Plateau make rainwater unable to gather there, reducing the breeding area and reproduction of mosquitoes [86]. Areas with high vegetation coverage are more conducive to mosquito breeding, and areas rich in water are more suitable for the reproduction of snails, thus promoting the spread of relevant vector-borne diseases.

Social regional factors, such as urbanization, have been identified as an important factor affecting vector-borne disease transmission. On the one hand, urbanization can destroy suitable environments for mosquito breeding, thus impacting the spread of relevant mosquito-borne diseases [80]. One the other hand, the level of urban facilities varies among urban, suburban, and rural. A study conducted in Brazil indicated that the risk of dengue was higher in more rural areas than in highly urbanized areas during extremely wet conditions, and the dengue risk following extreme drought was higher in areas that had a higher frequency of water supply shortages [94]. In addition, population structure also plays an important role since vector-borne diseases are more likely to spread in high-density population areas [97] and in different susceptible populations. For example, the high-risk group for malaria in Guangzhou is males aged 20–44 [98]; the unemployed and retired here were more likely to be infected with dengue [97]. Women, the elderly, and farmers in Shandong, Jiangsu, and Anhui are more susceptible to scrub typhus [50]; middle-aged and elderly farmers are high-risk populations for severe fever with thrombocytopenia syndrome [99], while the leishmaniasis tends to be transmitted in people under 20 years old in Xinjiang [79]. Additionally, agricultural ecosystems and artificial ecological engineering change the climate of local areas, leading to small-scale outbreaks of vector-borne diseases [78,100]. Socio-economic levels, public education levels, immunization rates, and people’s awareness of health can also play an important role in the transmission of vector-borne diseases [74,101,102].

However, these ecological and social regional factors are changing greatly. Climate change has a profound impact on the structure and function of natural ecosystems in China, such as severe land loss, change in vegetation cover, reduction in river runoff, and forest destruction, which interfere with vector habitats and alter the spread dynamics of pathogens across species. Moreover, China is in the rapid advance period of industrialization, urbanization, and modernization with a slowing total population growth, an aging population, and an adjusted fertility policy. The risk of vector-borne diseases will increase continuously in China in the future. However, at present, studies mostly focus on the effect of single meteorological factors on vector-borne diseases, although the outbreak of vector-borne diseases was proven to be determined by a combination of different meteorological factors. Only the studies on hemorrhagic fever with renal syndrome indicated the interaction and marginal effects among temperature, precipitation, humidity, and the effect of temperature and its interactions with relative humidity and rainfall on malaria [103]. It has been shown that the precipitation in the hottest season has a positive impact on the survival of mosquitoes [104], and warmer and wetter weathers will continuously occur in the future [105]. How these compound events affect vector-borne diseases, especially mosquito-borne diseases, is still unknown. Meanwhile, the studies mostly took the administrative regions as the research areas, which cannot account for the particular role of land type, vegetation cover, or water bodies, the ecological regional factors not distributed strictly according to administrative regions. These factors can greatly impact the living activities of vectors, playing in the effect path of climate factors on vector-borne diseases. Furthermore, with the background of various kinds of extreme climate events occurring frequently in China, we need know how extreme climate events affect vector-borne diseases. Lastly, it is important to figure out the role of social factors, which are of great significance for us learn more about the changing tendency of vector-borne diseases and prepare more to prevent them in the future.

In spite of our review of domestic climate and vector-borne diseases studies, this study has some limitations. First, we only considered articles after the year of 2000. Plague was once prevalent in China before the year of 2000. We may ignore the effect of climate on the rodent-borne diseases. Secondly, we only included studies on the Chinese mainland but not on Hongkong, Macao, and Taiwan, where are also high-risk areas for mosquito-borne diseases. This may make our overall understanding of vector-borne diseases not complete enough for the whole of China. Finally, we only included the studies with population health outcome, no research related to vectors included. This may not be conductive to comprehensively understanding the mechanisms of climate affecting vector-borne diseases.

## 5. Conclusions

This study systematically reviewed the research progress of the relationships between climate factors and vector-borne disease in mainland China. Mosquito-borne diseases have been identified as the most significant vector-borne diseases. Temperature, precipitation, and humidity were the main drivers contributing to the transmission of vector-borne diseases. The association between weather conditions and vector-borne diseases and the path of climate affecting vector-borne diseases showed vast differences between northern and southern China, which resulted from nature and social factors. In the future, studies on the effects of temperature and rainfall on mosquito-borne diseases, especially dengue fever, should be strengthened, as should the need to pay more attention to the effects of compound extreme events and the role of various nature and social factors at local levels.

## Figures and Tables

**Figure 1 biology-11-00370-f001:**
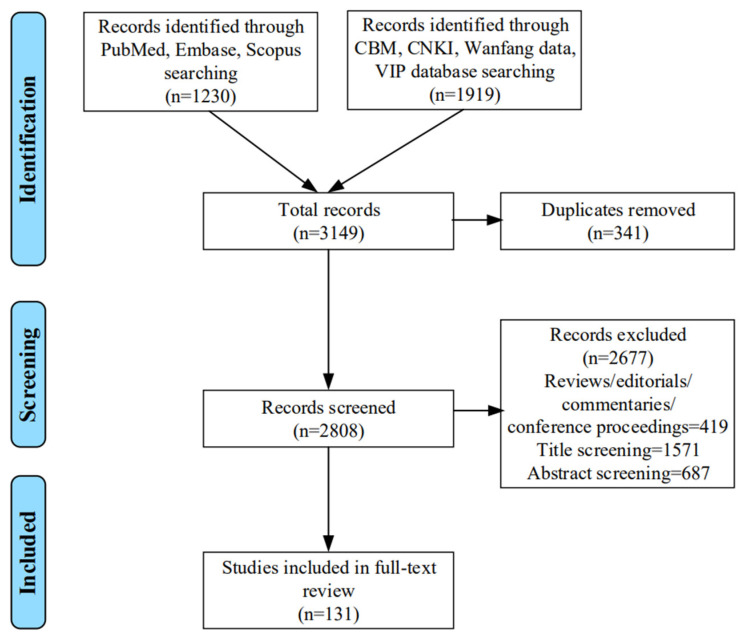
Flow chart of the selection process to retrieve all relevant studies.

**Figure 2 biology-11-00370-f002:**
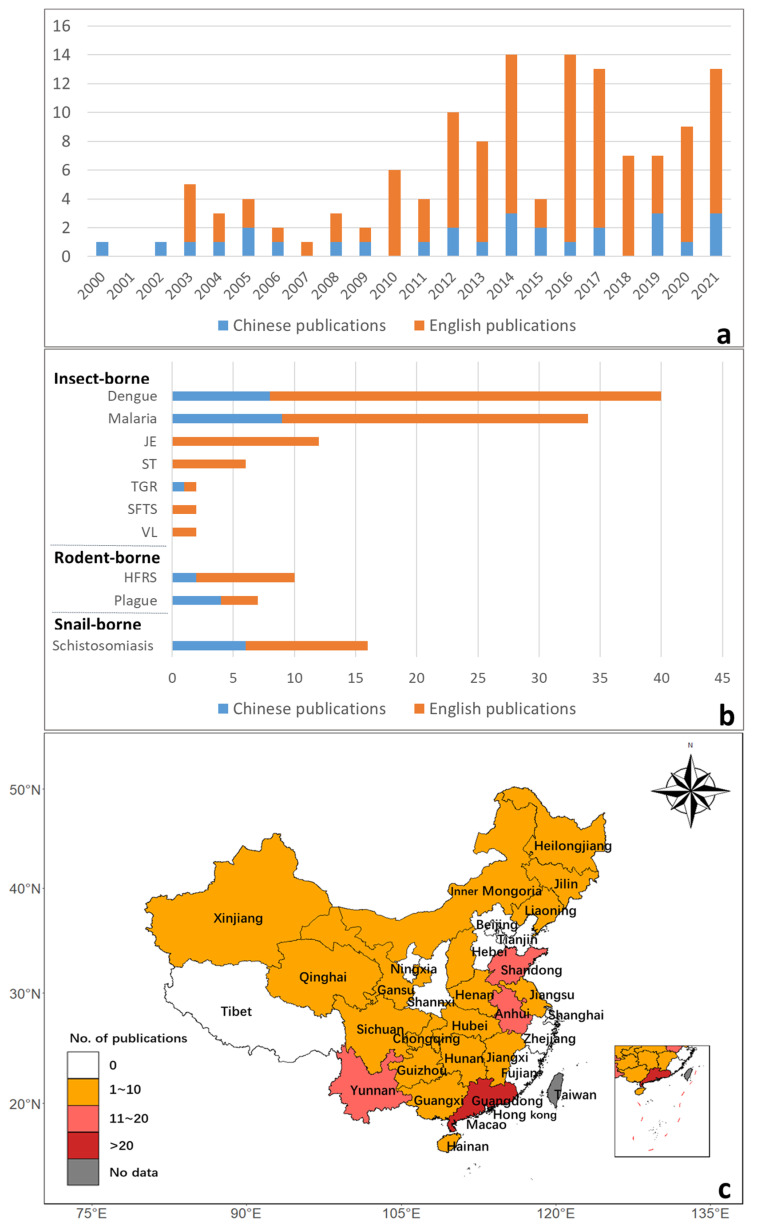
The number of reviewed publications during 2000–2021. (**a**) The temporal trend of publications on vector-borne diseases, JE (Japanese encephalitis), ST (scrub typhus), TGR (typhus group rickettsiosis), SFTS (severe fever with thrombocytopenia syndrome), VL (visceral leishmaniasis), HFRS (hemorrhagic fever with renal syndrome); (**b**) The number of studies on different vector-borne diseases; (**c**) The geographical distribution of the studies.

**Figure 3 biology-11-00370-f003:**
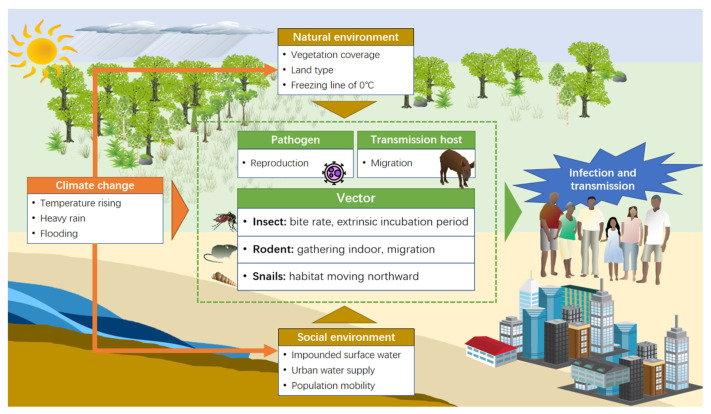
The main pathway of climate change impact on the risk of vector-borne diseases.

**Table 1 biology-11-00370-t001:** Summary characteristics of research on meteorological factors and vector-borne diseases in China.

Vector-Borne Disease	Vector	Study Area	Meteorological Factors	Outcome Metrics	Main Findings
Malaria	Mosquito	Shandong, Henan, Anhui, Jiangsu, Hubei, Sichuan, Chongqing, Guizhou, Yunnan, Guangdong, Hainan	Temperature, precipitation, humidity, air pressure, wind speed, sunshine, fog frequency, evaporation, flooding	Incidence, number of cases, detection rate	The association between meteorological factors and insect-borne diseases was nonlinear, consisting of reverse U-type and J-type shapes. The effects of rising temperature, rainfall, and humidity were beneficial to insect-borne disease transmission with lag effects. The correlations between wind speed, sunshine duration, air pressure, and insect-borne infectious diseases were negative. However, these correlations were different in some areas in China (see in Table 2).
Dengue	Mosquito	Guangdong, Fujian, Guangxi, Yunnan	Temperature, precipitation, humidity, air pressure, wind speed, sunshine, East Asian monsoon index, Southern Osmillation Index	Incidence, number of cases	
Japanese encephalitis	Mosquito	Shandong, Shaanxi, Hunan, Sichuan, Chongqing	Temperature, precipitation, humidity, air pressure, sunshine	Incidence, number of cases	
Scrub typhus	Mites	Shandong, Anhui, Jiangsu, Guangdong	Temperature, precipitation, humidity, air pressure, sunshine, wind speed, evaporation	Incidence, number of cases	
Typhus	Fleas	Liaoning, Yunnan	Temperature, precipitation, humidity	Number of cases	
SFTS	Ticks	Jiangsu	Temperature, humidity, wind speed	Incidence	
Leishmaniasis	Sandflies	Xinjiang	Temperature, precipitation, humidity	Number of cases	
Plague	Rodent	Gansu, Qinghai, Sichuan, Yunnan, Guizhou, Guangxi	Temperature, precipitation, humidity, Southern Oscillation Index, equatorial sea surface temperature in the eastern Pacific Ocean	Incidence, number of cases, bacteriological positive rate of plague, intensity of the outbreak, spread rate	The positive association between temperature, precipitation, and humidity and rodent-borne diseases was nonlinear with lag effects. Wind speed was negatively correlated with rodent-borne diseases. However, the results varied in different regions (see in Table 2).
HFRS	Rodent	Liaoning, Shandong, Anhui	Temperature, precipitation, humidity, air pressure, wind speed, sunshine, Southern Osm index	Incidence, number of cases	
Schistosomiasis	Snails	Anhui, Jiangsu, Jiangxi	Temperature, precipitation, humidity, sunshine	Incidence, infection rate, number of cases, acute schistosomiasis detectable rate	The association between schistosomiasis and temperature was negative, while the rainfall and humidity associations were positive. However, the results varied in different regions (see in Table 2).

**Table 2 biology-11-00370-t002:** The relationships between meteorological factors and vector-borne diseases according to the classification of different administrative regions in China.

Disease	Area	Time Period	Meteorological Factors		
			Temperature	Precipitation	Humidity
**Malaria**	**Shandong**				
	Jinan City	1959–1979	Max T (+) **	P (+)	H (+) *
			Min T (+) **		
	**Henan**				
	Yongcheng County	2006–2010	Monthly avg max T (+) ***	-	Monthly avg H (+) **
	**Anhui**	1990–2009	Monthly avg T(+) *	Monthly avg P (+) **	Monthly avg RH (+) *
	Shuchen County	1980–1991	Monthly avg max T (+) *** Monthly avg min T (+) ***	Monthly P (+) ***	Monthly avg RH (+) ***
	Hefei city	1999–2009	Monthly avg T (+) Monthly avg max T (+) *** Monthly avg min T (+) ***	P (+) *	H (+) ***
	Hefei City	1990–2011	Monthly min T (+) ***	P	RH (+) ***
	**Yunnan**				
	Mengla County	1971–1999	Monthly max T (+) * Monthly min T (+) *	Monthly P (−)	Monthly RH (−)
	125 counties	2012	Yearly avg T (+) **	Yearly P (+) **	
	**Guangdong**	2005–2013	High T (+)	P (J)	-
	Guangzhou city	2006–2012	Daily avg T (+) *	-	Daily RH (+) *
	**Hainan**	1995–2008	Monthly avg T (+) * Monthly avg max T (+) * Monthly min T (+) *	Monthly total P (+) *	-
**Dengue**	**Guangdong**				
	Guangzhou City	2006–2015	Extremely high T (+) *	Extremely high P (+) *	Extremely high H (+) *
	Guangzhou City	2005–2015	Monthly avg max T (+) **	Monthly total P (+) **	-
	Guangzhou City	2007–2012	Monthly avg T (+) **	-	Monthly avg RH (+) **
	Guangzhou City	2001–2006	Min T (+) ***	Monthly total P (+)	Min H (+)
	Guangzhou City	2000–2012	Monthly avg min T (+) *	Monthly total P (+) *	Monthly avg RH (+) *
	Guangzhou City	2005–2011	Daily avg T (+) * Daily min T (+) * Daily max T (−) *	Daily P (+)	Daily H (+)
	Zhongshan City	2001–2013	Monthly max T (+) * Monthly max DTR (+) *	-	Monthly avg RH (+) * Monthly max RH (+) *
	**Fujian**	1978–2017	Monthly avg T (+) *	Monthly total P (+) *	-
	**Guangxi**	1978–2017	Monthly avg T (+) *	Monthly total P (+) *	-
	**Yuanan**	1978–2017	Monthly avg T (+) *	Monthly total P (+) *	-
**Japanese encephalitis**	**Shandong**				
	Jinan City	1959–1979	Monthly avg max T (+) *** Monthly avg min T (+) ***	Monthly total P (+) *	Monthly avg RH (+) ***
	Linyi City	1956–2004	Monthly min T (+) **	-	Monthly avg RH (+) *
	**Shannxi**	2006–2014	Monthly min T (−)	Monthly P (+)	-
	**Anhui**				
	Jieshou County	1980–1996	Monthly avg max T (+) * Monthly avg min T (+) *	Monthly total P (+) **	-
	**Hunan**				
	Changsha city	2004–2009	Monthly avg max T (+) * Monthly avg min T (+) *	Monthly total P (+) *	Monthly avg AH (+) *
	**Sichuan**				
	Nanchong City	2007–2012	Daily avg T (+) *	-	Daily avg RH (+) *
	**Chongqin**				
	12 counties along the Yangtze River	1997–2008	Monthly avg T (+) ***	Monthly total P (−) ***	-
**Scrub typhus**	**Shandong**	2006–2013	Monthly avg T (reversed U) ***	Monthly total P (−) ***	Monthly avg RH (−) ***
	Laiwu City	2006–2012	Monthly avg T (+) **	Monthly avg P (+) **	Monthly avg RH (+) **
	**Anhui**	2006–2013	Monthly avg T (reversed U) ***	Monthly total P (−) ***	Monthly avg RH (+) ***
	**Jiangsu**	2006–2013	Monthly avg T (reversed U) ***	Monthly total P (−) ***	Monthly avg RH (+) ***
	Yancheng City	2005–2014	Monthly avg min T (+) ***	Monthly total P (+) ***	Monthly avg RH (−) ***
	**Guangdong**				
	Guangzhou City	2006–2012	Daily avg T (+) **	Daily P (+) **	Daily avg RH (−) *
**Typhus group rickettsiosis**	**Yunan**				
	Xishuangbanna	2005–2017	Weekly avg T (J) *	Weekly avg P (reversed U) *	-
**SFTS**	**Jiangsu**	2010–2016	Max T in warmest month (+) *	P in warmest month (+) *	-
**Leishmaniasis**	**Xinjiang**				
	Jiashi County	2005–2015	Monthly avg T (+) **	Monthly total P	Monthly avg RH (−) **
**Plague**	**Gansu**				
	Sunan County, Subei County	1973–2016	Monthly avg T (+) *	Monthly avg P (+) *	Monthly avg RH (−) *
	Yunnan	1982–2013	Extreme max T (−) **	-	Avg RH (+) **
**HFRS**	**Guizhou**	1982–2013	Extreme max T (−) **	-	Avg RH (+) **
	**Guangxi**	1982–2013	Extreme max T (−) **	-	Avg RH (+) **
	**Liaoning**	2005–2014	Weekly max T (+) *	Weekly P (+) *	Weekly avg RH (+) *
	Shenyang City	2004–2009	Monthly avg T (−) * Monthly avg max T (−) * Monthly avg min T (−) *	Monthly total P (−) *	Monthly avg RH (−) *
	**Heilongjiang**	2005–2014	Weekly max T (+) *	Weekly P (+) *	Weekly avg RH (+) *
	**Anhui**	2005–2014	Weekly max T (+) *	Weekly P (+) *	Weekly avg RH (+) *
**Schistosomiasis**	**Hubei**	1976–1989	Avg T in July (−) *	Avg P in July (−) *	-
	**Anhui**	1997–2010	Monthly avg T (−) *	Monthly total P (−) *	
	**Jiangxi**	2008	-	Monthly min P (−) ** Monthly max P (−) **	

The bold fonts in the region column are provinces in China. Max (maximum), Min (minimum), Avg (average), T (temperature), DTR (the difference between the maximum and the minimum daily temperature), P (precipitation), H (humidity), RH (relative humidity), AH (absolute humidity). “*” (The result is significant at level of α = 0.05), “**” (The result is significant at level of α = 0.01), “***” (The result is significant at level of α = 0.001).

## Data Availability

Not applicable.

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
