# Peer review of "Climate Change and Vector-Borne Diseases in China: A Review of Evidence and Implications for Risk Management"

_biology, 2022, doi:10.3390/biology11030370_

Round 1
Reviewer 1 Report
Title: Climate change and vector-borne diseases in China: A review of evidence and implications for risk management
Authors: Wu, Yurong and Huang, Cunrui
Overall: This is a literature review of Chinese research during the past 20 years focused on the intersection between vector-borne disease cases, climate change, landscape and social factors. The number of variables measured is comprehensive. Unfortunately the review is not cohesive and seems to be overly ambitious in attempting to cover literature from both the northern and southern regions. There are many generalizations the studies are not well-contextualized in the review.
The whole manuscript needs editing and some moderate grammatical work. Parts are OK, and parts are either somewhat repetitive or simply do not quite make sense.
A few notes to the authors:
- the word “research” in English is virtually always singular. Don’t use “researches”. This error is repeated throughout the manuscript. In English common use is “research studies”; for example line 148 in Results, “while dengue research studies were concentrated in the southern coastal area…” Or even "studies".
- Schistosomiasis is singular, e.g., line 151 in Results.
Abstract:
Line 19: “….the mechanism between them.” does not make sense in English. Revise please.
Here is an example of how the authors could edit a sentence:
Lines 26-31: Both the relationship and the mechanism*(s) show vast differences between northern and southern China resulting from landscape and social factors. We recommend that additional research be focused on the impact of these factors in particular on mosquito-borne diseases in the context of current and future climate change. Such information could be used to facilitate joint multi-sectorial prevention and control programs.
*Again, “mechanisms” line 26, is unclear. What do the authors mean?
I think the reduction of rodent-borne diseases is worthwhile mentioning in the abstract.
Introduction:
A good summary with a lot of up-to-date references. Some editing required.
Line 49: Replace “in our country” with “in China”. Here and throughout the manuscript.
Line 58: “….any steps as temperatures rise.”
Lines 66-69 – this sentence is particularly obscure. Revise.
Methods:
Line 126: Please list all vectors searched, rather than using “etc.”
Results:
In line 140-141, “The remaining 109 articles were included for full-text screening…” but in Figure 1, the number is 131. Explain this discrepancy.
Line 144, replace “our country” with China.
Lines 173-174. Spell out acronyms (such as ENMs, GEE…).
Line 178, what is “Qu”? I do not see this name in ref. 13.
Section 3.2: lines 174-176 mention the kinds of relationships (non-linear) and shape (U-shaped, J-shaped) but it is essential that the authors explain briefly what these mean.
Similarly, lines 200-201, “a dome-shaped relationship” needs to be contextualized.
Section 3.4: Lines 243-244, authors should briefly mention the differences between J- and U-shaped correlations.
Discussion:
This section is too long. Individual paragraphs are also overly long and some of the sentences are not linked to each other. It is hard to read and does not have a smooth narrative flow.
Conclusions:
This section should be revised to be more thoughtful. It is too similar to the Abstract.
References:
Use italics for all scientific names.
For example:
Ref. 14, Aedes
Ref. 38, Plasmodium vivax
Ref. 40, Marmota himalayana
Several additional references need to be modified for scientific names.
Also: Ref. 11, spell out C.c.c.o.C.M.
Ref. 12, add page numbers.
Tables:
Table 1 is fine, simply descriptive.
Table 2, which of these positive and negative correlations is/are statistically significant? At what level?
Figures:
Fine, useful.
Reviewer 2 Report
Dear Author/s,
Submitted manuscript Biology-1510326 “Climate change and vector-borne diseases in China: A review of evidence and implications for risk management”; is not presenting scientific intervention rather it is a database survey of the above-mentioned topic. Particularly, this survey has been done based on reports published in China and some Chinese databases. However, in discussion section, authors are talk about other parts of world for Zika virus and Lyme diseases. Overall, information is not concrete and have been pieced together. Some of the references cited here in the paper is not available on any search engine. This kind of region-specific survey would be helpful for public health policy maker of that country exclusively.
Biology is a broad-spectrum international journal and climate change is a global issue as well as vector-borne diseases are also prevalent everywhere. Information provided in this review is not sufficient discussed enough even in Chinese perspective like all causative factors, preventive measures, impact etc. Method has been unnecessarily elaborated. There is extensive need to work on English language of this manuscript. Authors has stated several places “Our Country”, although this review has been submitted to an international journal and follow up by readers throughout the world.
Eventually, there is a lot more efforts needed to make this report comprehensive and suitable for scientific reader.
All the best.
Reviewer 3 Report
This manuscript presents a comprehensive review of 131 studies on climate-sensitive mosquito-born diseases in China and surrounding countries. The review was conducted to gather data needed to help plan prevention programs to address the threat of climate-sensitive mosquito transmitted diseases in the country. The Review determined that: 1) temperature and humidity are the most concerned meteorological factors which can enhance transmission of vector borne diseases in China, and 2) both the relationship and mechanism show vast differences between the north and south in China due to numerous nature and social factors.
Round 2
Reviewer 1 Report
1) This resubmission is much improved but there are still some redundancies - for example, in (3. Results), the literature search description, lines 143-148 repeats what Figure 1 shows. I urge authors to reduce these lines to a single sentence that refers to Figure 1.
2) Line 156. Please cite a reference or explain why the number of studies increased significantly; also do not use the word "significantly" here unless a statistical analysis was conducted to compare numbers of pubs before and after this date.
3) Figure 2. The names of the main areas in Table 2 must be placed on the map (a better map too) - many readers will not know Chinese regions and if the authors want a broad readership that is also international, they need to make more of an effort.
4. There are still at least 50 minor edits needed, for example "severe" not "sever"; in places verbs are dropped or forgotten; e.g., line 232, "it would have adverse..." not "it would adverse"; line 236, "Studies" not "Stuides"; line 237, how far northward? give an estimate; line 265, replace "while mounted up in northern region" with "increased in the northern region".
5. The discussion repeats some of what the results state. It is really important to not repeat information especially what is already in the Tables and Figures. Just highlight the main trends.
Reviewer 2 Report
The Main drawback is that it is a review article and authors are trying to present/publish it as a Research article.
They are emphasising the methods for literature survey and discussion. Whereas this is not needed for review articles.
This review is very superficial with a limited spectrum for authors. Nothing new has been presented here which has been untouched before.
This article would be suitable for the International Journal of Environmental Research and Public Health ( IJERPH).
I am still firm with my previous decision. Please take the opinion of other reviewers.
Author Response
Please see in the attachment.
